

# Clinicopathological characteristics, treatment and survival of pulmonary large cell neuroendocrine carcinoma: a SEER population-based study

Ling Cao[1,*], Zhi-wen Li[2,*], Min Wang[3], Ting-ting Zhang[4], Bo Bao[1] and Yun-peng Liu[5]

[1] Department of Radiation Oncology, Cancer Hospital of Jilin Province, Changchun, Jilin, China
[2] Department of Anesthesiology, First Hospital of Jilin University, Changchun, Jilin, China
[3] Department of Pathology, Cancer Hospital of Jilin Province, Changchun, Jilin, China
[4] Department of Medical Oncology, Cancer Hospital of Jilin Province, Changchun, Jilin, China
[5] Department of Thoracic Surgery, First Hospital of Jilin University, Changchun, Jilin, China
* These authors contributed equally to this work.

Corresponding author
Yun-peng Liu,
liuyunpeng1848@163.com

## ABSTRACT

**Background:** This study was designed to investigate the clinicopathological characteristics, treatment and survival of patients with pulmonary large cell neuroendocrine carcinoma (LCNEC).

**Methods:** The Surveillance, Epidemiology and End Results database was utilized to identify patients diagnosed with pulmonary LCNEC between 2004 and 2013. Kaplan–Meier analysis was conducted to determine the overall survival (OS) and cancer-specific survival (CSS) rate. Univariate survival analysis along with log-rank test, and Cox proportional hazards model were employed to detect independent prognostic factors.

**Results:** Pulmonary LCNEC accounted for 0.58% (2972/510607) of the total number of lung and bronchus carcinoma. And a total of 1,530 eligible cases were identified, with the median follow-up time of 11 months. To be specific, the 3-, 5-year OS and CSS rates were 22.8%, 16.8% and 26.5%, 20.8% respectively. Generally, pulmonary LCNEC was commonly detected in the elderly (72.2%), males (55.9%), the upper lobe (62.0%) and advanced AJCC stage (65.5%). Multivariate analysis revealed that elderly [($\geq$60 and <80 years) HR:1.203, 95% CI [1.053–1.375], $P = 0.007$; ($\geq$80 years) HR:1.530, 95% CI [1.238–1.891], $P < 0.001$] and advanced AJCC stage [(stage III) HR:2.606, 95% CI [2.083–3.260], $P < 0.001$; (stage IV) HR:4.881, 95% CI [3.923–6.072], $P < 0.001$] were independent unfavorable prognostic factors, and that female (HR:0.845, 95% CI [0.754–0.947], $P = 0.004$)), surgery [(Segmentectomy/wedge resection) HR:0.526, 95% CI [0.413–0.669], $P < 0.001$; (Lobectomy/Bilobectomy) HR:0.357, 95% CI [0.290–0.440], $P < 0.001$; (Pneumonectomy) HR:0.491, 95% CI [0.355–0.679], $P < 0.001$] , chemotherapy (HR:0.442, 95% CI [0.389–0.503], $P < 0.001$) and radiation (HR:0.837, 95% CI [0.738–0.949], $P = 0.005$) were independent favorable prognostic factors.

**Conclusion:** To sum up, age at diagnosis, sex, AJCC 8th edition stage, surgery, chemotherapy and radiation were significantly associated with OS of patients with pulmonary LCNEC.

## INTRODUCTION

Pulmonary large cell neuroendocrine carcinoma (LCNEC) is an extremely rare disease with an incidence accounting for approximately 3% of all types of lung cancers (*Fasano et al., 2015*). LCNEC was first suggested by *Travis et al. (1991)* as a new type of solitary pulmonary neuroendocrine tumor, which was different from the typical, atypical carcinoid or small cell lung carcinoma (SCLC). In 1999 and 2004, the World Health Organization (WHO) admitted that LCNEC was a variant of large cell carcinoma, belonging to neuroendocrine tumors and one of the non-small cell lung cancer (*Rekhtman, 2010*; *Varlotto et al., 2011*).

Due to its low incidence and a lack of relevant clinical trial data, there is limited clinical understanding of its biological characteristics; hence, there are some controversies over the therapeutic strategies, especially in advanced patients (*Fasano et al., 2015*). Generally, most clinical studies of LCNEC are limited to retrospective studies using sample sizes less than 100 cases; the largest sample size published only included 127 cases (*Roesel et al., 2016*). In 2018, there was also a retrospective study concerning the prognosis of LCNEC by enrolling 126 patients (*Zhou et al., 2018*). However, these studies were all of small samples, with greatly varied results, therefore, it is necessary to conduct large-scale, multi-center clinical studies.

The Surveillance, Epidemiology and End Results (SEER) Program is supported by the National Cancer Institute (NCI). This Program contains research data of 18 different population-based cancer registries, which covers 30% of the United States population (*Duggan et al., 2016*). Hence, a number of studies on rare diseases have been performed by using large samples from the SEER database (*He et al., 2015*; *Sun et al., 2017*; *Embry et al., 2011*). However, studies on pulmonary LCNEC from SEER database have not been found. In consideration of a lack of published data on pulmonary LCNEC and the needs for clinical work, we performed an analysis of the SEER database to characteristics, prognosis and survival of patients with this disease.

## MATERIALS AND METHODS

### Ethics statement

The SEER Research Data Agreement was signed for accessing SEER information with the use of reference number 16462-Nov2016. Following approved guidelines, we performed the research methods to obtain data provided from the SEER database. The data analysis was considered by the Office for Human Research Protection to be non-human subjects who were researched by the United States Department of Health and Human Services, as they were publicly available and de-identified. Thus, it did not require any approval by the institutional review board.

### Study population

Patient data were obtained using the SEER database (Submission, November 2016). The SEER*State v8.3.5 tool, released on March 6, 2018, was used for determining and
selecting eligible patients. Additionally, the study duration ranged from January 1, 2004 to December 31, 2013 (because details about tumor size and extension were not available in the SEER database before 2004). The inclusion criteria were listed as follows: age at the diagnosis $\geq$ 20 years; LCNEC pathologically confirmed based on histology (ICD-O-3 8013/3); restriction on site recodes ICD-O-3/WHO 2008 (International Classification of Diseases for Oncology, Third Edition) to "Lung and Bronchus". The exclusion criteria were as follows: (1) patients younger than 20 years old; (2) patients with more than one primary cancer; (3) patients with missing or incomplete survival data; (4) patients without pathological confirmation based on histology; (5) patients with low-grade pathology (Grade I and Grade II) were excluded because LCNEC was a kind of high-grade neuroendocrine lung tumors; (6) patients without certain important clinicopathological information, including age, race, marital status, primary site, surgical type and AJCC stage. The remaining patients were defined as SEER primary cohort.

## Covariates

Covariates included the age at diagnosis (<60 years; $\geq$60 and <80 years; $\geq$80 years), gender (male; female), race/ethnicity (white; black; other), marital status (married; unmarried), primary site (main bronchus; upper lobe; middle lobe; main bronchus; lower lobe; overlapping lesion of lung), laterality (bilateral; left; right), differentiation (poorly differentiated, Grade III; undifferentiated, Grade IV; unknown), AJCC stage groups (8th edition) (I; II; III; IV), surgery (no surgery; segmentectomy/wedge resection; lobectomy/bilobectomy; pneumonectomy), chemotherapy (no/unknown; yes), radiation (no/unknown; yes). Overlapping was defined as cancer that extends over more than one lobe. The widowed or single (never married or having a domestic partner) or divorced or separated patients were classified as unmarried. Eighth edition stage group was then calculated for each patient according to tumor size, extension and seventh edition N/M stages.

The endpoints of this study were overall survival (OS), which was defined as the time interval from diagnosis to the most recent follow-up date, or date of death. Cancer-specific survival (CSS) was defined as the time interval from diagnosis to the most recent follow-up date or date of death caused by pulmonary LCNEC. There was a predetermined cut-off date based on the SEER 2016 submission database, containing death information until 2014. Therefore, the cut-off date was set at December 31, 2014 in this study.

## Statistical analysis

The Kaplan–Meier method was used to estimate the univariate analysis, along with log-rank test in order to assess the differences of OS stratified by each variable. Cox proportional hazards model was used to conduct multivariate survival analysis. SPSS software (SPSS Inc., Chicago, IL, USA, version 23) was used for statistical analysis, and GraphPad Prism 5 was used to generate the survival curve. A $P < 0.05$ was considered as statistical significance.

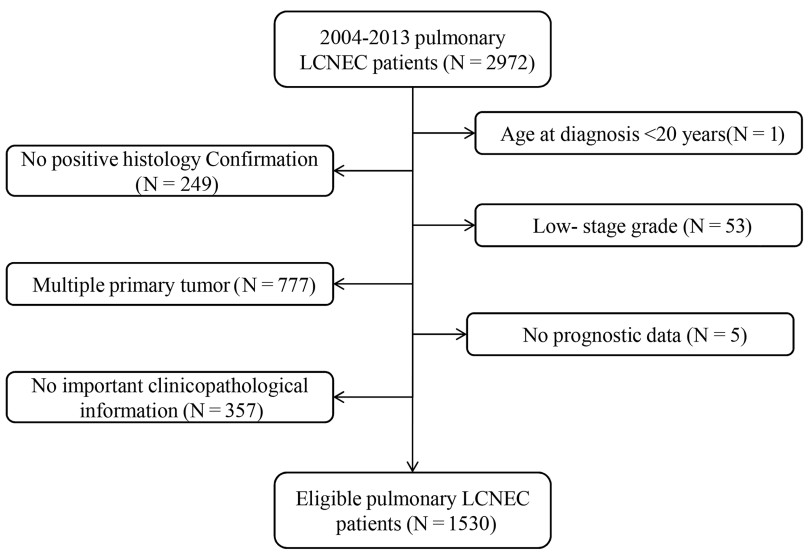

**Figure 1** **Flow chart for screening eligible patients.**     

## RESULTS

### Patient characteristics

Of the 510,607 patients with cancer in the lung and bronchus from 2004 to 2013, 2,972 (0.58%) were diagnosed with LCNEC. In total, 1,530 eligible patients were further enrolled in our research. The specific screening process was shown in Fig. 1. To be specific, median age at diagnosis was 66 years (27–92 years). Most of the patients were elderly people: 1,105 patients (72.2%) were ≥60 years old, while 142 patients (9.3%) were ≥ 80 years old. There were more male patients than female (55.9% vs. 44.1%). Only a small number of tumors (4.2%, $N = 64$) originated from the main bronchus, while most tumors occurred in the upper lobe of the lung (62.0%, $N = 948$). Stages III–IV were noted in 65.5% of patients ($N = 1,002$), while 25.8% ($N = 395$) and 8.7% ($N = 133$) were stages I and II, respectively.

Based on the available information, 40.7% of patients ($N = 622$) received cancer-directed surgery (CDS), including 133 patients receiving segmentectomy or wedge resection, 425 patients undergoing lobectomy or bilobectomy and 64 patients received pneumonectomy. In addition, 788 (51.5%) patients received chemotherapy and 565(36.9%) patients were treated with radiotherapy.

### Overall survival and prognostic factors

The median follow-up time of all eligible patients was 11 months (range 0–131 months). The three to five year OS and CSS rates were 22.8%, 16.8% and 26.5%, 20.8% respectively. The OS and CSS curves were shown in Fig. 2. In the univariate analyses, age ($P < 0.001$), sex ($P = 0.001$), primary site ($P < 0.001$) differentiation ($P < 0.001$), AJCC stage groups 8th edition ($P < 0.001$), surgery ($P < 0.001$), radiation ($P < 0.001$) were predictors of OS (Fig. 3). In addition, multivariate analysis further revealed that elderly [(≥60 and <80 years)HR:1.203, 95% CI [1.053–1.375], $P = 0.007$; (≥80 years) HR:1.530,

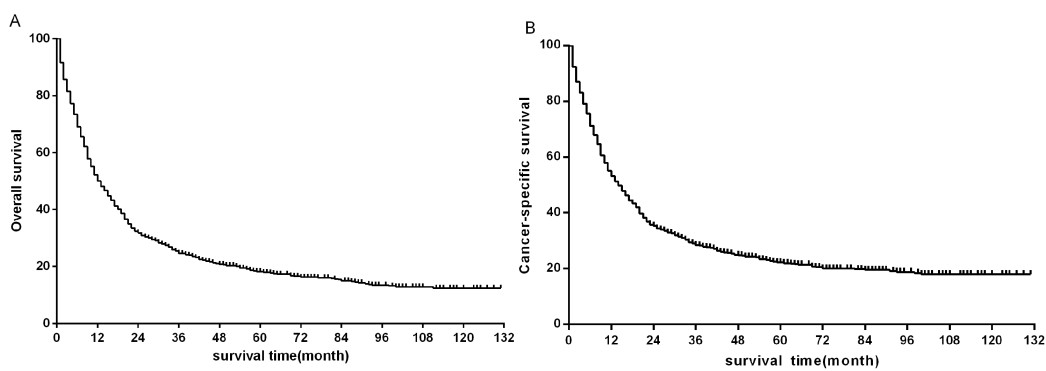

**Figure 2 Kaplan–Meier survival plots for eligible patients showing (A) overall survival (OS) and (B) disease-specific survival (DSS).**

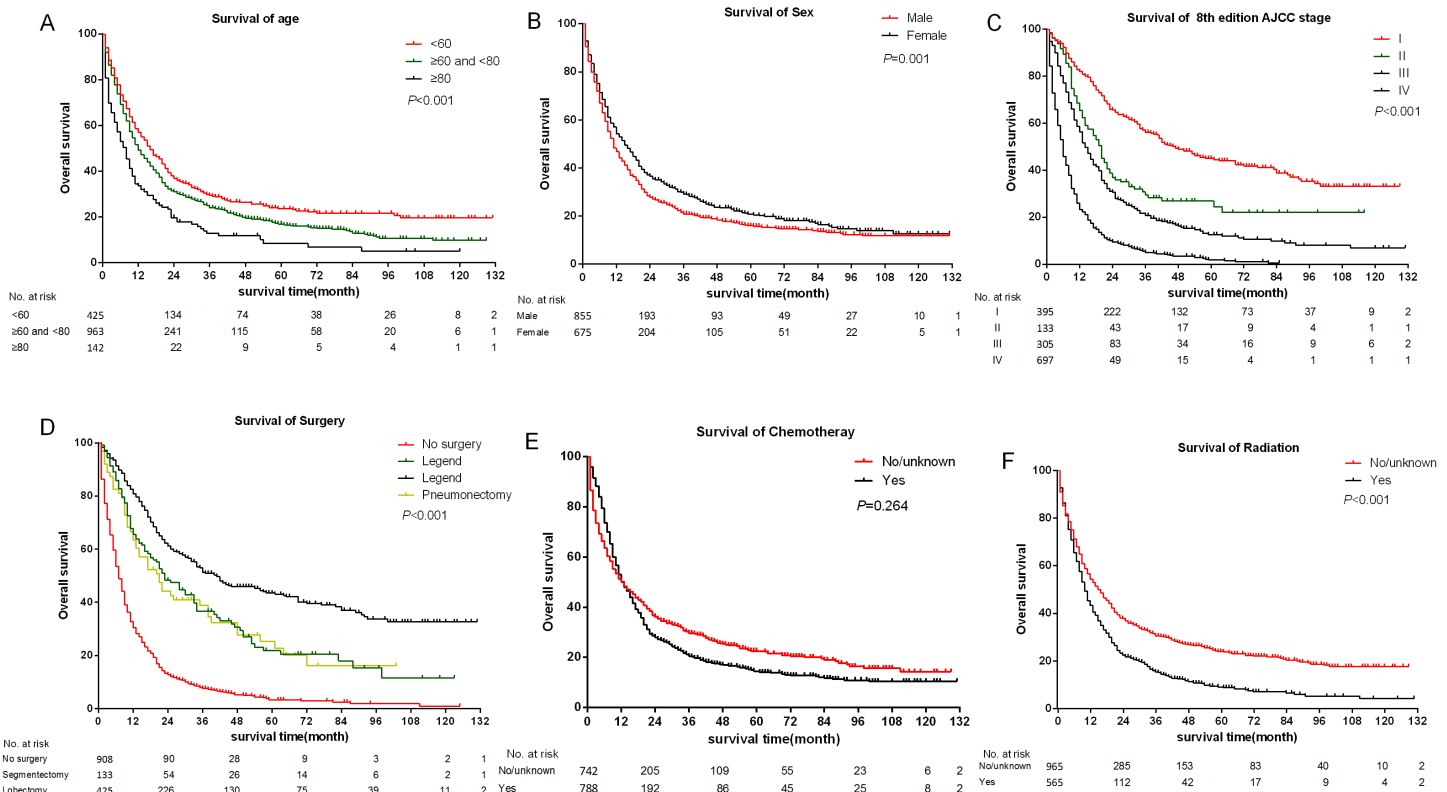

**Figure 3 Kaplan–Meier survival curves of some covariates without adjusted.** (A) Age at diagnosis; (B) sex; (C) summary stage; (D) surgery; (E) chemotherapy; (F) radiation.

95% CI [1.238–1.891], $P < 0.001$], advanced AJCC stage [(stage III) HR:2.606, 95% CI [2.083–3.260], $P < 0.001$; (stage IV) HR:4.881, 95% CI [3.923–6.072], $P < 0.001$] were independent unfavorable prognostic factors, and that female (HR:0.845, 95% CI [0.754–0.947], $P = 0.004$), surgery [(Segmentectomy/wedge resection) HR:0.526, 95% CI [0.413–0.669], $P < 0.001$; (Lobectomy/Bilobectomy) HR:0.357, 95% CI [0.290–0.440], $P < 0.001$; (Pneumonectomy) HR:0.491, 95% CI [0.355–0.679], $P < 0.001$], chemotherapy
(HR:0.442, 95% CI [0.389–0.503], $P < 0.001$) and radiation (HR:0.837, 95% CI [0.738–0.949], $P = 0.005$) were independent favorable prognostic factors (Table 1).

## DISCUSSION

Lung neuroendocrine tumors are considered as a heterogeneous sub-type from pulmonary cancers, accounting for approximately 20% of all lung cancers, which include typical, atypical carcinoid, SCLC and LCNEC. The occurrence of LCNEC consists of 2.1–3.5% of all lung cancers (*Fasano et al., 2015*). *Takei et al. (2002)* reported similar rate being at 3.1% (87/2,790). These findings were in accordance with *Iyoda et al. (2001)* who reported LCNEC a rate of 3.4% (72/2,070) in resected lung cancers. However, in our study, LCNEC accounted for only 0.58% of all lung cancers, which is considerably lower than previous reports. The main reason for this discrepancy is that previous research data were only based on post-operative data from a single center; however, our study is a multi-center study with a wide range of coverage, including both patients with surgery and without surgery. Therefore, the proportion of LCNEC in all lung cancer might be even lower than previously thought.

Unlike other lung neuroendocrine tumors, LCNECs was often associated with male sex, older age and cigarette consumption (*Sanchez de Cos Escuin, 2014*; *Selvaggi & Scagliotti, 2009*; *Asamura et al., 2006*). Furthermore, LCNEC was pathologically high-grade with poor prognosis (*Oshiro et al., 2004*). These clinicopathological features are also found in our study. In addition, our study demonstrated that most patients with pulmonary LCNEC were at an advanced stage when diagnosed.

Pulmonary LCNEC was biologically aggressive malignancies with a poor prognosis (*Iyoda et al., 2009*). The five-year OS after resection of LCNEC had been reported to range from 13 to 57% (*Varlotto et al., 2011*; *Younossian, Brundler & Totsch, 2002*; *Liang et al., 2015*). Prognosis was poor even in patients with potentially resectable stage I lung cancer, with five-year survival rates ranging from 27% to 67% (*Iyoda et al., 2009*). These previously reported results are consistent with our findings.

Because of a low incidence, there are few reports concerning the prognostic factors of pulmonary LCNEC. *Zhou et al. (2018)* found that tumor location, resection status, and EGFR mutational status were independent prognostic factors of LCNEC. In our study, we found that elderly, male and advanced stages are independent unfavorable prognostic factors and surgery, chemotherapy and radiation are protective factors for OS.

Primary surgery is the mainstream treatment for operable patients (*Fasano et al., 2015*), which also facilitates in accurate diagnosis (*Zacharias et al., 2003*). Our research found surgery as a positive independent prognostic factor, which further confirmed the importance of surgery in pulmonary LCNEC. However, most patients with pulmonary LCNEC were not eligible for surgical resection due to systemic or local tumor metastasis. Only 40.7% of patients could receive CDS.

If surgery is not possible, radiotherapy and chemotherapy can be considered. The role of chemotherapy and radiotherapy in the treatment of local or advanced pulmonary LCNEC is still unclear (*Hiroshima & Mino-Kenudson, 2017*). *Dresler et al. (1999)* reported no survival benefits from post-operative chemotherapy, radiation therapy or both in

**Table 1 Univariate and multivariate analyses of overall survival (OS) in the eligible patients.**

| Variables | N (%) | Univariate analysis | Multivariate analysis | |
|---|---|---|---|---|
| | | *P*-value | HR [95%CI] | *P*-value |
| Age | | <0.001 | | <0.001 |
| <60 | 425 (27.8%) | | Reference | |
| ≥60 and <80 | 963 (62.9%) | | 1.203 [1.053–1.375] | 0.007 |
| ≥80 | 142 (9.3%) | | 1.530 [1.238–1.891] | <0.001 |
| Sex | | 0.001 | | 0.004 |
| Male | 855 (55.9%) | | Reference | |
| Female | 675 (44.1%) | | 0.845 [0.754–0.947] | |
| Race | | 0.885 | | NI |
| White | 1283 (83.9%) | | | |
| Black | 187 (12.2%) | | | |
| Other# | 60 (3.9%) | | | |
| Marital status | | 0.276 | | NI |
| Married | 839 (54.8%) | | | |
| Unmarried | 691 (45.2%) | | | |
| Laterality | | 0.289 | | NI |
| Bilateral | 4 (0.4%) | | | |
| Left | 623 (40.3%) | | | |
| Right | 903 (59.3%) | | | |
| Primary site | | <0.001 | | 0.443 |
| Main bronchus | 64 (4.2%) | | Reference | |
| Upper lobe | 948 (62.0%) | | 0.917 [0.700–1.202] | 0.531 |
| Middle lobe | 82 (5.4%) | | 1.118 [0.793–1.577] | 0.524 |
| Lower lobe | 408 (26.7%) | | 0.891 [0.673–1.181] | 0.423 |
| Overlapping lesion | 28 (1.8%) | | 0,836 [0.507–1.377] | 0.482 |
| Differentiation | | <0.001 | | 0.322 |
| Grade III | 586 (38.3%) | | Reference | |
| Grade IV | 191 (12.5%) | | 1.157 [0.957–1.399] | 0.133 |
| Unknown | 753 (49.2%) | | 1.035 [0.909–1.179] | 0.603 |
| AJCC stage groups 8th edition | | <0.001 | | <0.001 |
| I | 395 (25.8%) | | Reference | |
| II | 133 (8.7%) | | 2.619 [2.027–3.384] | <0.001 |
| III | 305 (19.9%) | | 2.606 [2.083–3.260] | <0.001 |
| IV | 697 (45.6%) | | 4.881 [3.923–6.072] | <0.001 |
| Surgery | | <0.001 | | <0.001 |
| No surgery | 908 (59.3%) | | Reference | |
| Segmentectomy/wedge resection | 133 (8.7%) | | 0.526 [0.413–0.669] | <0.001 |
| Lobectomy/Bilobectomy | 425 (27.8%) | | 0.357 [0.290–0.440] | <0.001 |
| Pneumonectomy | 64 (4.2%) | | 0.491 [0.355–0.679] | <0.001 |

(Continued)

| Variables | N (%) | Univariate analysis | Multivariate analysis | |
|---|---|---|---|---|
| | | P-value | HR [95%CI] | P-value |
| Chemotheray | | 0.264 | | <0.001 |
| No/unknown | 742 (48.5%) | | Reference | |
| Yes | 788 (51.5%) | | 0.442 [0.389–0.503] | |
| Radiation | | <0.001 | | 0.005 |
| No/unknown | 965 (63.1%) | | Reference | |
| Yes | 565 (36.9%) | | 0.837 [0.738–0.949] | |

**Notes:**
N, number; HR, hazard ratio; 95% CI, 95% confidence index; NI, not included in the multivariate survival analysis.
[#] Other: American Indian/AK Native, Asian/Pacific Islander.

patients with resected LCNEC. *Shimada et al. (2012)* found that the overall response rate to initial chemotherapy or chemoradiotherapy and the survival outcomes of high-grade neuroendocrine carcinoma—probable LCNEC were comparable to those of SCLC. In our study, we found that chemotherapy and radiotherapy were independently protective factors of pulmonary LCNEC.

Currently, the NCCN treatment guidelines suggest using the same approach as with other NSCLC tumors (*Wood et al., 2018*). Although, some researchers have found that treatment similar to SCLC was more appropriate than NSCLC for advanced LCNEC (*Sun et al., 2012*). Analysis of K-ras-2, p53 and C-raf-1 expressions has indicated pulmonary LCNEC are more genetically similar with SCLC, compared with NSCLC (*Przygodzki et al., 1996*). Furthermore, SCLC and pulmonary LCNEC are categorized as high-grade neuroendocrine tumors, with similar clinical and histopathological characteristics. In addition, there were an increased number of scholars doubting if pulmonary LCNEC treatments being all similar to NSCLC was the right approach (*Fernandez & Battafarano, 2006*).

To our knowledge, our study is the largest retrospective analysis of LCNEC until now, with a total number of 1,530 patients. This sample data are extracted from the SEER database, which is considered to be accurate and reliable. Through our research, physicians could have a better understanding of the clinicopathological features, survival and the treatment of patients with pulmonary LCNEC. However, there are certain limitations in our study. Although 11 variables were involved, there are still some important variables that SEER does not includ, such as the chemotherapy regiments, blood tests, molecular test information and surgical margin status. Besides, selection bias might exist as we only included patients with complete information of involved variables.

## CONCLUSIONS

In conclusion, our study finds that LCNEC has a higher incidence in elderly people, males and advanced AJCC stage. In addition, elderly (≥60 years), male and advanced AJCC stages are independent unfavorable prognostic factors, while surgery, chemotherapy and radiation are independent protective factors of OS.

### Funding

This work is supported by a grant (to Zhi-wen Li) from Fundament Research Funds for Central Universities (451170306057). The funders had no role in study design, data collection and analysis, decision to publish or preparation of the manuscript.

### Grant Disclosure

The following grant information was disclosed by the authors:
Fundament Research Funds for Central Universities: 451170306057.

### Competing Interests

The authors declare that they have no competing interests.

### Author Contributions

- Ling Cao conceived and designed the experiments, performed the experiments, analyzed the data, contributed reagents/materials/analysis tools, prepared figures and/or tables, authored or reviewed drafts of the paper, approved the final draft.
- Zhi-wen Li performed the experiments, authored or reviewed drafts of the paper, approved the final draft.
- Min Wang approved the final draft.
- Ting-ting Zhang approved the final draft.
- Bo Bao approved the final draft.
- Yun-peng Liu analyzed the data, contributed reagents/materials/analysis tools, approved the final draft, design.

### Data Availability

The raw measurements are available in Supplementary File 1.

### Supplemental Information

Supplemental information for this article can be found online at http://dx.doi.org/10.7717/peerj.6539#supplemental-information.

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
