# Peer review of "Clinicopathological characteristics, treatment and survival of pulmonary large cell neuroendocrine carcinoma: a SEER population-based study"

_PeerJ, doi:10.7717/peerj.6539_

## Round 0.1 · original submission · Major Revisions

There are several major and minor concerns with the manuscript submitted. If authors decide to resubmit, please address all issues raised by the three reviews and provide a point by point response as to how each issue was addressed in the revised manuscript.

Reviewer 1 ·

Basic reporting

203: please fix the typo"respectable"
214: The authors keep referring that there is no standard treatment for pulmonary LCNEC. This is not true. There is no standard treatment for metastatic LCNEC and oncologists often use either NSCLC or SCLC chemotherapy regimens but early-stage pulmonary LCNEC is treated akin to NSCLC. Surgery and chemoradiation are standard treatments for Stage I-III pulmonary LCNEC.
229: Authors state that surgery and radiation are the favorable prognostic factor. This is expected since its the early stage disease that receive either surgery or chemoradiation and hence the prognosis is better. This needs to be mentioned or elaborated in the discussion. Surgery and radiation are in a way surrogates for early-stage disease in this analysis, but since authors did not analyze TNM staging this data is unavailable.
196: authors state that majority pulmonary LCNEC patients were either east coast or west/pacific coast residents and were Caucasians. This needs to be clarified since coastal areas are where the majority of the population lives. Was this data adjusted for population density? Similarly, the majority of the US population is Caucasian, can we get a comment on the incidence of LCNEC based on the race which is adjusted for population?

Experimental design

Experimental design is good.

Validity of the findings

Analysis is straightforward. Authors have used appropriate statistical tools.

Additional comments

The grammar and flow of article can, in general, be improved. Would recommend getting the manuscript proofread by a native english speaker.

Will compliment authors for taking up this project, review of pulmonary LCNEC epidemiological data is important in spreading awareness about this rare disease and to help foster therapeutic interventions. Will definitely advocate in favor of accepting this manuscript for publication.

Reviewer 2 ·

Basic reporting

no comment

Experimental design

no comment

Validity of the findings

no comment

Additional comments

In this manuscript, the Authors aim to investigate the clinical pathological characteristics, treatment and survival of pulmonary large cell neuroendocrine carcinoma. Using the SEER database data on 2556 patients were obtained. They found that LCNEC have a higher incidence in elderly people, males, late stage and peripheral localization. They also identified 9 variables that can be easily obtained in multivariate analyses.
1. The SEER is very poor database to use as the clinical detail in this administrative database is somewhat poor. Many factors are lacking which can only be obtained in institutional databases.
2. Just using “surgery” as the predictive variable does not well inform clinical practice. What type of surgery needs to be specified.
3. TNM staging is a very important prognostic factor for the prognosis of patients with LCNEC, however, the author did not take it into account.
4.The manuscript would likely benefit from copy edits since there are several misused and misspelled words. Attached Non-Native Speakers of English Editing Certificate should be attached.
5.The authors did not clarify how they dealt with missing data, particularly for the predictive variables.
6.Please report 3 and 5-year CSS.
7.Table 2: Please also report HRs for univariate analysis. How HRs were calculated? This was not reported in the manuscript.
8.Methods. There are 3 registries subgroups (SEER9, SEER13, SEER18). It is not clear to me the specific database used by Authors. ICD-O-3 codes used for SEER database query should be specified in a supplementary table, along with their explanation.

Reviewer 3 ·

Basic reporting

(1) The total survival curve was not provided.
(2) Table 2 is redundant since most information can be found in Figure 2.
(3) The survival time in Figure 2 should better be with an interval of 12 months and patients at risk should be added.
(4) There are some grammatical errors, such as in line 84, “This program contain” should be “contains”.

Experimental design

(1) The authors investigated patients with LCNEC between 2000 and 2013 in SEER databases. However, SEER database recorded their data according to different modes as years went by. For example, the 6th and 7th TNM stages were provided in records after 2004 and 2010, respectively. The authors can restage the patients into the newest 8th TNM stage manually by carefully checking the records in recent years. They should better analyze the prognostic significance of T, N, M, and TNM stage instead of the old stage classification (local, regional, and distant).
(2) The chemotherapy status, a quite important variable associated with prognosis, is also now provided in SEER database. The authors can visit the website (https://seer.cancer.gov/data/treatment.html), obtain the chemotherapy status and use it in the survival analyses.
(3) Differentiation, marital status, and insurance were also frequently reported to be correlated with the prognosis, but not included in the analyses. The left or right lobes were not analyzed, either.
(4) It is not appropriate that age only was divided into <60 and ≥60. Patients at the age of 80 couldn’t have a similar prognosis as those at the age of 60. The authors should better classify the age with an interval of 10 years.
(5) The authors declared that 80.7% LNCEC patients were with peripheral localization, which is incorrect. The tumor was located in the upper, middle, or lower lobe did not mean it was peripheral. Peripheral lung cancer refers to lung cancer derived from respiratory bronchioles.

Validity of the findings

(1) The authors should better validate their results using their own records or data of other public databases.

Additional comments

In this manuscript, Dr. Gao et al. investigated the clinicopathological features, treatment, and survival of patients with lung large cell neuroendocrine carcinoma (LCNEC) using SEER data. They found that the survival of LCNEC was not well, and LCNEC was associated with elderly, male, peripheral localization, and so on. After univariate and multivariate analyses, the authors found that age, sex, stage, surgery, and radiation were independent prognostic factors of LCNEC.
There are several concerns as mentioned above.

---

## Round 0.2 · Major Revisions

Although the manuscript has been modified and improved, there are still issues remaining that need to be addressed. Please address these issues and provide a point by point response regarding how each issue is addressed in the revised manuscript

Reviewer 3 ·

Basic reporting

no comment

Experimental design

no comment

Validity of the findings

no comment

Additional comments

The revised manuscript is an improvement of the previous one. However, there are still some major concerns.
(1) The TNM stages are very important prognostic variables, which is also mentioned by other reviewers. The authors can manually restage the stages using the variables CS tumor size, CS extension, CS lymph nodes, CS mets at dx, CS site-specific factor 1, and CS site-specific factor 2 recorded in SEER database. The details can be found on the website (https://seer.cancer.gov/archive/manuals/2016/appendixc.html).
(2) The factors not significant in univariate analyses, such as race (p = 0.501), marital status (p = 0.215), laterality (p = 0.237), and chemotherapy (p = 0.796), should not be included in the multivariate analysis.

---

## Round 0.3 · accepted · Accept

The manuscript has now been satisfactorily revised and accepted for publication.

Reviewer 1 ·

Basic reporting

This is a revision; manuscript is well written. No issues

Experimental design

Methodology is appropriate.

Validity of the findings

Data is consistent with what should be expected.

Additional comments

Revisions are accurate and adequate. I have two minor suggestions.

1# I would recommend avoiding vague terms like "elderly" for e.g. "Most of the patients were elderly people"

2# Authors report a low incidence 0.5% iof all pulmonary tumors as compared to single institutional series where in it is reported in about 2-3% of thoracic tumors. Authors suggests that "The main reason for this discrepancy is that previous research data were only based on post-operative data from a single center; however, our study is a multicenter study with a wide range of coverage, including both patients with surgery and without surgery. Therefore, the proportion of LCNEC in all lung cancer might be even lower than previously thought." This explanation might not be completely accurate. Pulmonary LCNEC diagnosis often needs a good quality biopsy or post op specimen. Inadequate FNAC or cytology can often miss the diagnosis. So I will recomend that authors should give both explanations and rather than stating that LCNEC incidence is low because SEER database analysis suggest so. They should also add that the incidence could be low in SEER due to underreporting/misdiagnosis in SEER.

Reviewer 3 ·

Basic reporting

no comment

Experimental design

no comment

Validity of the findings

no comment

Additional comments

it's now worth to be accepted now.